# Help-seeking behaviour and attitudes towards internet-administered psychological support among adolescent and young adults previously treated for cancer during childhood: protocol for a survey and embedded qualitative interview study in Sweden

Joanne Woodford ,[1] Jenny Månberg,[2] Åsa Cajander,[3] Pia Enebrink,[4] Arja Harila-Saari,[5] Josefin Hagström,[1] Mathilda Karlsson,[1] Hanna Placid Solimena,[6] Louise von Essen[1]

For numbered affiliations see end of article.

**Correspondence to**
Dr Joanne Woodford;
joanne.woodford@kbh.uu.se

## ABSTRACT

**Introduction** A subgroup of adolescent and young adult childhood cancer survivors (AYACCS) are at increased risk of psychological distress. Despite this, AYACCS experience difficulties accessing psychological support. E-mental health (e-MH) may offer a solution to reduce this treatment gap. However, research examining e-MH for AYACCS has experienced difficulties with recruitment, retention and adherence. Such difficulties may relate to: (1) help-seeking behaviour and/or (2) e-MH acceptability. The overall study aims are to: (1) examine potential associations between health service use factors, informed by Andersen's behavioural model of health services use, and help-seeking behaviour; (2) examine attitudes towards e-MH interventions; and (3) explore perceived need for mental health support; past experience of receiving mental health support; preferences for support; and barriers and facilitators to help-seeking.

**Methods and analysis** An online and paper-based cross-sectional self-report survey (98 items) and embedded qualitative interview study across Sweden, with a target sample size of n=365. Participants are aged 16–39 years, diagnosed with cancer when 0–18 years and have completed successful cancer treatment. The survey examines sociodemographic and clinical characteristics, actual help-seeking behaviour, attitudes towards e-MH, stigma of mental illness, mental health literacy, social support and current symptoms of depression, anxiety, and stress. Survey respondents with past and/or current experience of mental health difficulties are invited into the qualitative interview study to explore: (1) perceived need for mental health support; (2) past experience of receiving mental health support; (3) preferences for support; and (4) barriers and facilitators to help-seeking. Potential associations between health service use factors and help-seeking behaviour are examined using univariable and multivariable logistic regressions. Qualitative interviews are analysed using content analysis.

## Strengths and limitations of this study

► The mixed methods design may aid a rich and insightful understanding of the investigated phenomena.

► Results may be used to inform the development of acceptable, relevant, and accessible mental health interventions for adolescent and young adult childhood cancer survivors.

► Reasons for not receiving a mental health intervention are not explored and only those who report receiving support in the past 6 months are defined as help-seeking.

► This approach may exclude adolescent and young adult childhood cancer survivors who have attempted to seek formal help (help-seeking) but did not receive help.

► No conclusions concerning cause and effect can be drawn given the cross-sectional design.

**Ethics and dissemination** Ethical approval has been obtained from the Swedish Ethical Review Authority (Dnr: 2020-06271). Results will be disseminated in scientific publications and academic conference presentations.
**Trial registration number** ISRCTN70570236.

## INTRODUCTION

Medical improvements have resulted in an overall 5-year survival rate of 81.2% from cancer diagnosed during childhood (0–18 years) in Northern Europe.[1] However, significant challenges, such as late effects, remain after treatment completion.[2][3] Survivors of childhood cancer report a number of long-term adverse physical and developmental

outcomes[4–6] and social and economic impacts, including poorer educational and occupational achievements.[7 8] A further common late effect is symptoms of psychological distress, for example, post-traumatic stress, anxiety and depression, symptoms of which have been found to persist decades postdiagnosis.[9–11] Difficulties with psychological distress are further complicated given that adolescence is a period of increased vulnerability to mental health problems,[12] with difficulties such as depression and anxiety commonly first appearing before 24 years of age.[13] Indeed, individuals diagnosed with cancer during childhood, adolescence and young adulthood have been found to be at an increased likelihood of being prescribed hypnotics and anxiolytics[14 15] of using antidepressant medication over the long-term,[16] and are at an elevated risk of suicide.[17] As such, adolescent and young adult childhood cancer survivors (AYACCS), commonly defined as cancer survivors diagnosed during childhood currently aged between 15 and 39 years,[18] are particularly vulnerable to experiencing adverse psychological and social outcomes, and there is a significant need for long-term care and support.[5] However, despite a clear need for the provision of mental health support, a mental health treatment gap remains, with few AYACCS being offered professional psychological support.[19–22] Reasons for this treatment gap may pertain to a lack of qualified healthcare professionals, poor service provision, geographical barriers[23] and public stigma (eg, the general public holding negative beliefs about cancer survivors, eg, being less competent).[24]

E-mental health (e-MH), defined as the provision of psychological support via information technology and new media (eg, online, smartphone applications, video conferencing),[25] may represent a solution to reduce the mental health treatment gap.[26] One example of e-MH is internet-administered cognitive behavioural therapy (ICBT). ICBT has been found to be effective for a range of common mental health difficulties, such as depression and anxiety,[27] including for young adult populations[28] and has been found to be as effective as traditional face-to-face CBT.[29] e-MH interventions may help overcome practical barriers associated with accessing traditional face-to-face psychological support (eg, geographical, financial and time-related barriers)[30] that have been observed within the AYACCS population.[31] Furthermore, e-MH is associated with anonymity and privacy[32] thus potentially overcoming barriers associated with public stigma. Indeed, research has identified that the provision of e-MH interventions may be a potential solution for the AYACCS population.[33–38] However, research examining e-MH support for an AYACCS population has encountered challenges with recruitment, attrition, and internet use.[34 38] For example, one study examining the feasibility and acceptability of e-MH recruited only 28 participants over a 12-month period and experienced almost 30% attrition.[38] Another feasibility study of e-MH for adolescents and young adults who had experienced cancer during the adolescent period recruited only 6 participants

from a total of 320 potential participants invited into the study, and experienced a 100% attrition rate.[34] Indeed, a recent review of e-Health interventions for AYACCS concluded that engaging the population was challenging, and there is currently a lack of literature concerning how to better engage the population.[36] However, challenges with recruitment, attrition and internet use are common in the field of e-MH research.[39–41] Potentially, challenges with recruitment, attrition and internet use may be related to help-seeking behaviour and the acceptability of e-MH for the population.

## Help-seeking behaviour

Low rates of formal help-seeking behaviour for mental health difficulties is common among an adolescent and young adult population in general.[42] A number of potential barriers to help-seeking have been identified, including stigma, embarrassment, and poor mental health literacy (MHL).[43 44] AYACCS report difficulties with stigma and alienation in relation to the cancer disease[45] and as such may be reluctant to identify themselves as experiencing mental health difficulties. Furthermore, while social support has been found to facilitate help-seeking behaviour for mental health difficulties in adolescent and young adult populations,[43] AYACCS commonly report difficulties with social withdrawal and loneliness.[46 47] However, at present there is a lack of research concerning help-seeking behaviour for mental health difficulties experienced by an AYACCS population.

A frequently used approach for identifying factors associated with help-seeking is Andersen's behavioural model of health services use behaviour.[48] The model hypothesises a number of factors to be associated with help-seeking behaviour: (1) predisposing factors including sociodemographic variables (eg, age and gender), and health-related beliefs and knowledge[49]; (2) enabling factors relating to logistical elements to receiving care and resources (eg, financial factors and social support)[50]; (3) external environmental factors (eg, location); and (4) need-related factors (eg, symptoms of psychological distress). Recently, the model has been adopted to examine factors associated with accessing both traditional and e-MH services within a young adult (aged 18–39 years) population.[32] Interestingly, the study identified a weaker utilisation of e-MH by young people who had experienced negative childhood experiences during childhood (eg, family conflict, bullying or lack of affection).[32] This finding is of particular importance seeing that negative experiences of childhood are normally associated with increased mental healthcare utilisation.[51] Furthermore, given evidence to suggest some AYACCS may experience bullying related to their cancer experience[52] and poor family functioning for example conflict,[53] it may be important to examine whether negative childhood experiences are associated with reduced mental healthcare utilisation in an AYACCS population. Furthermore, to the best of our knowledge, Andersen's behavioural model of health services use has not yet been applied to an AYACCS population and

factors associated with help-seeking in the population remain unclear.

## Acceptability

While adolescents appear to consider e-MH interventions to be potentially acceptable, they still report a preference for face-to-face over e-MH interventions.[54] Indeed, there is some indication e-MH resources are not widely accessed by young people.[55] One potential difficulty may be related to what has been coined the '*technology push'*, whereby e-MH solutions are designed to fit available technology, rather than to meet the unmet needs and preferences of end-users.[56] Furthermore, research indicates poor uptake and ongoing use of e-MH interventions that have been implemented into healthcare settings within the general population.[57] As such, there is a need to understand preferences and attitudes towards e-MH interventions within specific populations, via both quantitative and qualitative research, to inform future e-MH developments.[58] To the best of our knowledge, there has been no exploration to date of the attitudes and preferences towards e-MH interventions within an AYACCS population. In addition, tailoring e-MH interventions to specific patient groups has been associated with increased effectiveness[59] and adherence.[60] Rates of attrition may be reduced, and adherence increased, if the perspective of the population using the intervention is taken into account when developing e-MH interventions.[61] Furthermore, research into the psychological needs of an AYACCS population indicates a need for increased focus on the specific psychological and psychosocial needs of the population to enable to provision of more tailored support to the cancer experience.[62] As such, there is need to explore AYACCS general preferences for psychological support, including potential intervention content.

## Study aims and research questions

The overall aims of the study are to: (1) examine potential associations between health service use factors, informed by Andersen's behavioural model of health services use, and help-seeking behaviour; (2) examine attitudes towards e-MH interventions; and (3) explore perceived need and past experiences of mental health support, preferences for support, and barriers and facilitators to help-seeking. Specifically, the following research questions will be examined:

1. Are there associations between predisposing, enabling, environmental, and needs-related health service use factors, informed by Andersen's behavioural model of health, and help-seeking behaviour within an AYACCS population?
2. What attitudes are held by an AYACCS population towards e-MH interventions?
3. What is the perceived need for mental health support within an AYACCS population?
4. What are AYACCS experiences of past receipt of cancer-specific and non-cancer-specific mental health support?

5. What preferences do an AYACCS population hold towards mental health support related to their cancer experience?
6. What are the barriers and facilitators do an AYACCS population experience seeking help for mental health support related to their cancer experience?

## METHOD AND ANALYSIS
### Study design

An online and paper-based, cross-sectional, self-report survey and embedded qualitative interview study across Sweden. The study has been registered in the ISRCTN clinical trial registry (ISRCTN70570236).

### Eligibility criteria

Eligible participants will: (1) be an adolescent or young adult, aged 16–39 years, at study start; (2) have been diagnosed with childhood cancer when 0–18 years; (3) have completed successful cancer treatment (including relapses) at a minimum 3 months ago according to self-report; (4) be able to read and write in Swedish; and (5) currently reside in Sweden.

### Recruitment

A multifaceted approach to recruitment will be adopted, using two main strategies.

#### Swedish Childhood Cancer Registry

Personal identification numbers of AYACCS will be obtained from the Swedish Childhood Cancer Registry (National Quality Registry, initiated in 1982). Next, addresses will be obtained via NAVET, a population registry held by the Swedish Tax Agency. NAVET is continuously updated with important information about the Swedish population, including deaths and can, as such, provide current addresses and information concerning the vital status of AYACCS. The Swedish Tax Agency will provide this information via a secure online file transfer system (e-transport) on the same day information is requested by the study team. Study invitation letters will be sent to AYACCS on the same day that information is provided via e-transport in order to minimise the likelihood of inviting a deceased AYACCS. AYACCS will be invited to participate by the study team on a weekly basis, using blocks of 100 until the target sample size has been reached. Blocks will be selected randomly, with minimisation used to ensure balance between factors that may affect outcomes: sex (male; female); current age (16–24 years; 25–39 years); and age at diagnosis (0–12 years; 13–18 years).

Study invitation packs sent to home addresses will include a study invitation letter, study information, a link to the online survey via the U-CARE portal (www.u-care. se), a secure internet research platform to support data collection and provision of e-MH interventions,[63] and a study log-in code. All potential participants identified via the Swedish Childhood Cancer Registry and invited via mail-out will be provided with a link to the online survey on the U-CARE portal and a study log-in code. In addition,

given evidence to suggest the provision of choice between web-based and paper-based surveys can increase survey response rates among childhood cancer survivors,[64] we will include a paper-based survey and freepost envelope with each invitation letter.

## Technology-based approaches

Considering that response rates to mail-out surveys have declined over the past two decades,[65] we will also use technology-based recruitment approaches shown to be effective.[66] Indeed, social media recruitment is successful in recruiting young adult cancer survivor (aged 20–35 years) populations.[67] Specifically, we will identify community-based organisations and support groups with an online presence for AYACCS in Sweden. Examples may include: the Swedish Cancer Foundation and Childhood Cancer Foundation and Young Cancer. Organisations and support groups will be asked to place advertisements on their websites; distribute advertisements via e-mail lists and e-newsletters; and post advertisements on social networking sites (eg, Facebook, Twitter, and Instagram).

Each online advertisement will include brief study information, a link to the online survey on the U-CARE-portal, and a study log-in code unique to each organisation and technologically based recruitment type to allow for the examination of success of each technology-based recruitment strategy. Organisations and support groups using social media will be encouraged to re-post the advertisement at regular intervals following evidence suggesting that re-posting is required to meet recruitment targets.[68]

## Sample size estimation

The Swedish Childhood Cancer Registry has some 9880 Swedish persons diagnosed when 0–18 years registered since 1982. However, some will have been classified as having a benign tumour, some have been registered twice and some will be deceased. As such, the number of potentially eligible cases will be lower and estimated to be approximately 5745. Sample size calculation indicated that a minimum of 365 participants would be required, with a power of 0.9 and $p<0.05$ (two-tailed). Assuming a response rate of approximately 40%, as indicated by a systematic review of participation rates in self-administered questionnaire studies with childhood cancer survivors,[69] we estimate inviting 913 potential participants to achieve the required sample size.

## Procedure
### Online survey

Potential participants accessing the survey via the U-CARE-portal will be presented with study information in both text and video format. Participants will not be offered any financial reimbursements for taking part in the survey. Those interested will be able to log into the U-CARE portal using a study log-in code and will need to identify themselves via mobile telephone and/or Bank ID (a citizen authentication system). Potential participants will then be able to provide informed consent online and

will be presented with brief screening questions asking for: (1) date of birth; (2) age when diagnosed with cancer; (3) date of last cancer treatment completed; (4) able to read and write in Swedish (yes/no); and (5) whether currently residing in Sweden (yes/no). Eligible participants will be provided access to the online survey (see 'Survey items' section). The online survey will be open for a period of 6 weeks. Each item completed in the survey is auto-saved and participants are able to select 'save and continue later' if they wish to return to the survey before submitting. On completion of the survey, a purposive sample of participants with past and/or present experience of mental health difficulties will be invited to take part in an interview (see 'Embedded qualitative interview study' section for further details). These participants will be provided with study information and online consent form, as well as asked to provide contact details.

### Paper-based survey

Potential participants receiving study invitation via mail-out will be able to complete a paper-based survey. The survey includes a written consent form and brief screening questions as the online survey. Participants can return the paper-based survey to the study team using the freepost envelope provided in the study invitation pack. In addition, participants will be asked to provide consent if they agree to be contacted about participation in an interview study.

### Reminders

Reminder study invitation packs will be resent at 2 and 6 weeks after sending the initial study invitation to AYACCS who do respond to the study invitation. It will be clearly stated in the study information sheet that reminder letters will be sent to non-responders. In addition, participants who consent to complete the online survey will receive reminders via SMS and email to complete the survey at 4 weeks, 2 weeks and 2 days prior to the survey closing.

### Survey items

An online and paper-based survey, consisting of 98 items in Swedish. The survey comprises eight subsections: (1) sociodemographic characteristics (11 items); (2) clinical cancer-related characteristics (3 items); (3) actual help-seeking behaviour (2 items); (4) attitudes towards internet-administered interventions (17 items); (5) stigma of mental illness (5 items); (6) MHL (26 items); (7) social support (12 items); and (8) current symptoms of depression, anxiety and stress (21 items). One open question at the end of the survey offers and opportunity to provide any further information concerning emotional distress and preferences for support after childhood cancer. Potential predisposing, enabling, environmental, and need-related factors examined with the survey are summarised in table 1.

### Sociodemographic characteristics

Data on the following background and sociodemographic characteristics will be collected: (1) sex; (2) relationship

**Table 1** Factors to be examined as per Andersen's behavioural model of health services use

| Andersen's behavioural model of health services use | Factors* |
|---|---|
| Predisposing | Sex (male/female/other)<br>Age (16–24/25–39)<br>Time since first cancer treatment (years)<br>Place of birth (born in Sweden/born outside of Sweden)<br>Childhood negative events (yes/no)<br>Level of education (≤upper secondary school/>upper secondary school)<br>Public stigma (Stigma Scale for Receiving Psychological Help)<br>Mental health literacy |
| Enabling | Type of cancer (leukaemia/central nervous system tumour/solid tumour/lymphoma/other)<br>Relationship status<br>Employment status (yes/no/studying)<br>Social support (12-item Interpersonal Support Evaluation List) |
| Need-related | Symptoms of depression (DASS-21 depression subscale)<br>Symptoms of anxiety (DASS-21 anxiety subscale)<br>Symptoms of stress (DASS-21 stress subscale)<br>Self-reported experience of mental health difficulties in past 6 months (yes/no) |
| Environmental | Region of Sweden (south/mid/north)<br>Rural or urban area |
| Actual help-seeking behaviour (dependent variable) | Receipt of mental health support in the past 6 months<br>Type of mental health support received in the past 6 months (formal/technology/informal) |

*Outcomes selected informed by previous research.[32 42 50 70]
DASS-21, Depression Anxiety and Stress Scale.

status; (3) having children or not; (4) Swedish county of residence; (5) living in urban or rural area; (6) level of education; (7) employment status; (8) place of birth; (9) mother's place of birth; (10) father's place of birth; and (11) childhood negative events (bullying, family conflict, lack of affection).

### Clinical cancer-related characteristics

The following clinical characteristics related to cancer will be collected: (1) type of cancer (leukaemia, central nervous system tumour, solid tumour, lymphoma or other malignancy); (2) number of relapses (0, 1 or >1); and (3) current late effects of cancer (multiple choice listed).

### Actual help-seeking behaviour

The following items will be used to examine actual help-seeking behaviour: (1) emotional health difficulties related to the childhood cancer experience; (2) the type of support received by those who accessed support in the past 6 months and/or in the past (not including the past 6 months): (a) face-to-face individual or group support from mental health professional; (b) face-to-face support from a general practitioner/family doctor; (c) technology-administered mental health support (eg, online therapy programme, smartphone application, video conferencing or teleconferencing); (d) technology-administered support from a general practitioner/family doctor (eg, online, video conferencing or teleconferencing); (e) social media support group; (f) phone help-line; (g) family, partner or friend/s; (h) religious leader; and (i) other. Type of support will be

categorised into: (1) face-to-face professional support; (2) technology-administered professional support; and (3) informal support (social media support group, family, partner, friends and religious leader). Those who sought support for mental health difficulties in the past 6 months and received support from a health professional either face-to-face or technology assisted will be defined as 'help-seeking'.[70]

### Attitudes towards internet-administered interventions

The 17-item e-Therapy Attitudes Measure (ETAM)[71 72] will be adopted to measure attitudes towards internet-administered interventions, however, for the purpose of the present study one item concerning health insurance companies was excluded due to not being relevant in the Swedish context. The questionnaire begins with definitions of three types of internet-administered interventions: (1) unguided internet-administered self-help programmes; (2) therapist-guided internet-administered self-help programmes; and (3) psychological therapy delivered via video conferencing. Subsequently, respondents are required to state which type of internet-administered intervention they would prefer, or state if they would not use internet-administered treatment at all. Subsequently, the ETAM is presented and explores attitudes towards internet-administered interventions. Specifically, respondents are instructed to state whether they agree with each statement on a 5-point rating scale ranging from 0 ('strongly disagree') to 4 ('strongly agree') and are asked to rate items considering therapist-guided

internet-administered self-help programmes. The ETAM has been demonstrated to have good internal-consistency reliability (α=0.89).[72]

## Symptoms of depression, anxiety and stress
The Swedish version[73] of the (7-item depression, 7-item anxiety and 7-item stress subscales) Depression Anxiety and Stress Scale (DASS-21)[74 75] will be adopted to examine symptoms of depression, anxiety and stress. The DASS-21 has adequate psychometric properties[76] and satisfactory test–retest reliability,[77] with support for the factor structure and convergent validity of the Swedish version.[73]

## Stigma of mental illness
The 5-item Stigma Scale for Receiving Psychological Help (SSRPH)[78] will be adopted to measure public stigma, that is, negative perceptions concerning the receipt of mental health services. The measure has demonstrated to be reliable and valid in studies with young adults[79] and adolescents.[80] Respondents rate degrees of agreement using a 4-point Likert scale ranging from 0 ('strongly disagree') to 3 ('strongly agree'), with a higher score representing a higher level of stigma.

## Social support
The 12-item Interpersonal Support Evaluation List (ISEL-12)[81] will be adopted to examine perceptions of social support. Responses are provided on a 4-point Likert scale, ranging from 1 ('definitely false') to 4 ('definitely true').

## Mental health literacy
A 26-item multicomponent MHL measure, demonstrated to have good internal consistency (of 0.83 of the KR-20 coefficient=0.83),[82] will be adopted to assess knowledge-oriented, beliefs-oriented and resource-oriented MHL. The first 22 items examining knowledge-orientated and beliefs-orientated MHL are answered on a 5-point Likert scale ranging from 0 ('strongly disagree') to 4 ('strongly agree') and the option of "I don't know". The response format for the final four items, examining resource-orientated MHL, is 'yes' or 'no'.

## Translation of English language outcome measurements into Swedish
Currently, there are no Swedish versions of the following outcome measurements: (1) ETAM, (3) SSRPH, (4) ISEL, and (5) MLH. The following steps will be taken to translate: (1) one clinical psychologist, native Swedish speaker with advanced knowledge of English, will translate each outcome measurement to Swedish and (2) translations will be reviewed, discussed and revised with a research assistant in our study team who is a native Swedish speaker with advanced knowledge of English. Professional back translation to English and discussion with the original author is out of scope for the present study due to time and resource limitations. However, permission to translate outcome measurements to Swedish will be sought.

## Embedded qualitative interview study
An embedded qualitative interview study will be conducted to explore: (1) perceived need for mental health support; (2) past experience of receiving mental health support; (3) preferences for support; and (4) barriers and facilitators to help-seeking. As this study will be examining mental health help-seeking, only respondents meeting the following criteria will be invited into the study: (1) self-reported current and/or past experience of mental health difficulties during life time (yes) and/or (2) DASS-21 depression score ≥5 and/or (3) DASS-21 anxiety score ≥4 and/or DASS-21 stress score ≥8. Respondents will be provided with full study information and an online consent form. Respondents meeting these criteria will be invited consecutively until thematic data saturation is met.[83 84] As the decision concerning whether data saturation has been met is made during the analytic process, it is difficult to determine the number of interviews a priori; however, we may anticipate interviewing approximately 20–30 participants.[85] As we are interested in both barriers and facilitators to seeking help, we will endeavour to interview participants classified as help-seekers and non-help-seekers. For participants who send the consent form and reply slip via the post, those meeting the inclusion criteria will be contacted by the study team to provide full study information and obtain informed consent. Interviews will be conducted either via the telephone or a secure video conferencing system on the U-CARE portal. Interviews will be audio recorded with informed consent and transcribed with personal information removed. Participants will be informed about the recording and give approval when giving consent to the interview. An interview guide will be developed, consisting of open-ended questions, structured around the third research objective covering the following topics: (1) perceived need for mental health support; (2) past experience of receiving mental health support (formal/informal, cancer-specific/non-cancer-specific); (3) preferences for support (eg, type of support, emotional difficulties to be target and intervention content areas); and (4) barriers and facilitators to help-seeking behaviour (eg, perceived attitudes of peers, knowledge/information, economic resources, geographical location, stigma). The interview guide will be partially informed by emerging survey results and findings from studies examining help-seeking behaviour in young adults.[32 42 50 70] Interviews are estimated to last 45–90 min.

## Data analysis
### Statistical analysis
All statistical analyses will be conducted using SPSS V.26.0 software (IBM SPSS Statistics for Windows, 2019, Armonk, New York, USA). Sociodemographic and clinical characteristics will be presented with descriptive statistics. The potential association between predicting factors as informed by Andersen's behavioural model of health services use (predisposing, enabling, environmental, and need-related factors) and the dependent variable help-seeking behaviour, for example, receipt

of mental health support in the past 6 months will be examined using univariable and multivariable logistic regressions.[70] The non-help-seeking group will used as the reference group.[70] First, univariable analyses will be conducted for each predictor separately. Variance inflation factor will be used to examine multicollinearity prior to the regression analysis. A multivariable logistic regression will be performed using stepwise manual backward selection procedure. All predictor variables will initially be included, and non-significant predictors will be eliminated with only variables significantly related to the dependent variable ($p<0.05$) retained. For both the univariable and multivariable logistic regressions, ORs will be presented, alongside 95% CIs, and p values. Given the heterogeneous nature of the study population, subgroup analysis will be performed by stratifying the population by age and sex. The same method of modelling will be used for the subgroup analyses for age (15–19 years vs 20–29 years vs 30–39 years) and sex (male vs female). The subgroup analysis will be exploratory and with results used to inform subsequent studies.

## Qualitative analysis

Audio recordings will be transcribed verbatim. NVivo V.12 software (NVivo qualitative data analysis software, 2018; QSR International) will be used to assist data analysis. An inductive content analysis approach[86] will be adopted to analyse transcriptions. Only manifest content will be analysed and the following steps will be undertaken: (1) each interview transcript will be read multiple times; (2) meaning units will be identified; (3) meaning units will be labelled using descriptive codes; and (4) codes will be sorted into themes and subthemes.[86] To ensure trustworthiness,[87] the following strategies will be adopted: (1) two members of the research team, one researcher with experience of qualitative research and one research assistant trained in qualitative research will analyse each interview separately with any discrepancies in analysis discussed; (2) discussion of emerging themes with the wider study team; (3) audit trails; (4) triangulation; and (5) disconfirming case analysis.[88] Furthermore, extracts of data supporting each theme and subtheme will be presented to further improve the transparency of the analysis.[89]

## Patient and public involvement

The protocol was developed without patient and public involvement (PPI). However, we will seek to involve AYACCS in pilot-testing the survey and in data interpretation of the content analysis of the embedded semi-structured interviews. Specifically, a consultation approach to PPI will be adopted[90] in pilot-testing the survey and a panel of AYACCS research partners will be asked to complete the survey (online and paper based) and comment on the feasibility, acceptability, and relevancy of the survey. Furthermore, results of the content analysis will be presented to the panel to explore whether themes identified by the research team reflect their own experiences. The perspective of the AYACCS research partners will be incorporated into the interpretation of results and their involvement will be reported in accordance with Guidance for Reporting Involvement of Patients and the Public-short form.[91]

## DISCUSSION

To the best of our knowledge, this study will be the first study to examine mental health help-seeking behaviour and attitudes towards e-MH support in an AYACCS population. Results will provide an understanding of: (1) potential associations between predisposing, enabling, environmental, needs-related health service use factors, and help-seeking behaviour; (2) attitudes towards e-MH; and (3) perceived need and preferences for mental health support, alongside barriers and facilitators to help seeking in the population. Our study design allows us to collect both qualitative and quantitative data and therefore will provide a more in-depth and rich understanding of mental health help-seeking behaviour and attitudes towards e-MH support in an AYACCS population. The identification of factors both positively and negatively associated with help-seeking behaviour may help identify potential targets to improve help-seeking behaviour in the future. For example, interventions could be developed to target and overcome factors found to be negatively associated with help-seeking[92] and subsequently enhance help-seeking behaviours in an AYACCS population in the future. Furthermore, increasing our understanding of attitudes towards e-MH support and preferences regarding the provision of mental health support in the population may inform the development of more acceptable and relevant interventions for this underserved population. In addition, examining preferences for support will enable the development of future psychological interventions that are specifically tailored to and target the needs on an AYACCS population seeking support in relation to their experience of cancer.

Despite the strengths of this study, the design presents some limitations. First, only those AYACCS who report receiving mental health support in the past 6 months are defined as help-seeking. However, this approach may lead to the exclusion of AYACCS who have attempted to seek formal help but not accessed. Second, the cross-sectional design does not allow us to draw any conclusions regarding cause and effect. Third, the study does not offer any compensation to participants. Research suggests compensation can facilitate the recruitment of minority populations and those from lower socioeconomic backgrounds.[93] Subsequently, this may limit the representativeness of the sample and generalisability of results, especially to marginalised young adults who may already be at an elevated risk of not seeking mental health support.[94] Forth, the current study includes a broad age range (16–39 years) and it may be expected that adolescents will have differing needs and preferences to young adults in their 20s or 30s. Indeed, a common limitation of research in the area is the inclusion of heterogeneous

samples across different age groups.[95] However, the study is designed to be a first step towards developing more acceptable and relevant interventions for the population and may inform future research with specific AYACCS subgroups, taking into account a number of sociodemographic and clinical factors (eg, gender, current age, age at diagnosis, and cancer type). Fifth, the length of the survey may lead to respondent fatigue (eg, lower levels of attention and motivation in later sections of the questionnaire), potentially resulting in poorer quality data or missing data. Finally, we will not evaluate response bias via comparisons of respondents versus non-responders for mailed surveys.

In conclusion, results of the study may have the potential to improve access to tailored, relevant, and acceptable mental health support for this currently underserved population.

## ETHICS AND DISSEMINATION

The study is approved by the Swedish Ethical Review Authority (Dnr: 2020-06271). The rights and welfare of participants will be ensured by all research being conducted in accordance with the Helsinki Declaration. Informed consent will be collected, ensuring participants are aware of requirements for study participation. Participants will be free to withdraw from the study at any time, without providing reason. Contact details for the principal investigator (coauthor LvE), the U-CARE Health and Safety Officer and Uppsala University Data Protection Officer will be provided to all participants should there be any cause for concern regarding the conduct of the study.

All data will be handled according to the Patient Data Act (2008:355) and General Data Protection Regulation (EU 2016/679). Data collected via the online survey via the U-CARE portal will be securely stored on Uppsala University servers. Data collected via the paper-based survey will be entered into an Access Database, stored on Uppsala University servers. Paper questionnaires will be stored in locked secure filing cabinets, only accessible by authorised members of the study team. Participant contact details for the interview study will be stored separately on a USB stick in a locked secure filing cabinet, separate from study data. Interviews will be audio recorded, with audio files uploaded onto and stored on Uppsala University secure server, using participant identifier numbers and immediately deleted from devices. Interview transcripts will omit any personally identifiable data and will be stored on a secure server.

Survey results will be reported in line with the Checklist for Reporting Results of Internet e-Surveys[96] and results of the embedded qualitative interview study will be reported in accordance with the Standards for Reporting Qualitative Research.[97] Results will be published in scientific publications in peer reviewed journals and conference presentations. Lay language summaries will also be provided to all community-based organisations and support groups who support recruitment into the study.

**Author affiliations**
[1]Clinical Psychology in Healthcare, Department of Women's and Children's Health, Uppsala University, Uppsala, Sweden
[2]Child and Adolescent Psychiatry, Region Vasternorrland, Sundsvall, Sweden
[3]Department of Information Technology, Uppsala University, Uppsala, Sweden
[4]Department of Clinical Neuroscience, Karolinska Institute, Stockholm, Sweden
[5]Pediatric Oncology, Department of Women's and Children's Health, Uppsala University, Uppsala, Sweden
[6]International Maternal and Child Health Care, Department of Women's and Children's Health, Uppsala University, Uppsala, Sweden

**Contributors** JW: conceptualisation, methodology, writing—original draft, supervision, project administration, funding acquisition; JM: methodology, resources, writing—original draft, project administration; ÅC: writing—review and editing; PE: writing—review and editing; AH-S: writing—review and editing; JH: resources, writing—review and editing, project administration; MK: methodology, resources, writing—review and editing, project administration; HPS: resources, writing—review and editing, project administration; LvE: conceptualisation, methodology, writing—review and editing, supervision, funding acquisition.

**Funding** This work is supported by U-CARE, which is a Strategic Research environment funded by the Swedish Research Council (Dnr 2009-1093) and Fredrik O Ingrid Thurings Stiftelse (grant number 2018-00437).

**Competing interests** None declared.

**Patient and public involvement** Patients and/or the public were not involved in the design, or conduct, or reporting, or dissemination plans of this research.

**Patient consent for publication** Not required.

**Provenance and peer review** Not commissioned; externally peer reviewed.

**ORCID iD**
Joanne Woodford http://orcid.org/0000-0001-5062-6798

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
