## [Reviewer comments · BMJ Open]

ARTICLE DETAILS

TITLE (PROVISIONAL)	Help-seeking behaviour and attitudes towards internet-administered psychological support among adolescent and young adults previously treated for cancer during childhood: Protocol for a survey and embedded qualitative interview study in Sweden
AUTHORS	Woodford, Joanne; Månberg, Jenny; Cajander, Åsa; Enebrink, Pia; Harila-Saari, Arja; Hagström, Josefin; Karlsson, Mathilda; Placid Solimena, Hanna; von Essen, Louise

VERSION 1 – REVIEW

REVIEWER	Ursula Sansom-Daly UNSW Sydney, Australia
REVIEW RETURNED	11-Aug-2020

GENERAL COMMENTS	Thank you for the opportunity to review the protocol for this interesting study. Overall the protocol is well written and I believe the study has potential to answer some important questions in AYA survivorship care and guide the development of more targeted, appropriate models of mental health care for AYA survivors of childhood cancer. I have a few comments/queries I hope will serve to strengthen this protocol. MAJOR COMMENTS - My major concern about this study/protocol as written is regarding the wide range of the sample – it seems that young people aged 16-39 now can participate, provided they were 0-18 years at time of diagnosis, and minimum 3 months post-treatment. So, it would seem that a young person who was an infant when they were diagnosed and is 39 now would still be eligible? This concerns me from a few perspectives: 1) I imagine that many young people particularly <5 or so years at diagnosis will not recall much of their cancer experiences; 2) you have a very large group in age who will each have experienced the disruption of cancer at different developmental stages, ranging from pre-school through to being treated for cancer during the end of high school. My questions related to these issues are as follows: o 1. How are the questions phrased relating to ‘emotional distress experienced during cancer treatment’, and how will you account for young people who cannot recall this?o 2. Assuming that you are simply treating their cancer experience as a risk factor that may increase their need for mental health support (but not the reason for their wanting to seek support per se – i.e., not assuming that they want support specifically to address psychological issues associated with their cancer experiences) – how will you tease this apart in your questions and subsequent analyses? You’re your questions about childhood negative events ask about whether these occurred chronologically prior to their
---

cancer diagnosis (as a history of e.g., school bullying prior to cancer diagnosis, for example, would predispose them to poorer social functioning/social anxiety during the treatment period, which would in turn likely exacerbate their mental health symptoms during/after treatment)? Also, will any of your items assess what issues they would want to seek help for, and whether this has been impacted in any way by their cancer experiences?

o 3. Given your aims are to inform practice, how do you intend to use your analyses to do this with such a broad group? Is it the case in Sweden that the same group/hospital setting/community cancer support organisation would provide psychological support to all 16-39 year olds with some kind of history of cancer? It seems that the clinical implications for program development for teenagers with a recent history of cancer (e.g., <5 years) may be different than young adults in their 20s and 30s who were diagnosed with cancer as small children...

- Your survey seems very long – 98 items is quite extensive, and I would be worried about the feasibility of this and the level of burden in completing it. A few questions: 1) will participants be reimbursed at all for their time in doing the survey? If so, you need to clarify this; if not, I think this needs to be acknowledged as a limitation as I think the type of participants who are willing and able to complete such a lengthy survey for no compensation will not necessarily be representative as a group. The commitment and general self-organisation required to complete such a long survey may also preclude some people from participating precisely for the reasons that they also may have trouble organising themselves to seek mental health support, so I think this isn't irrelevant to your core study question actually. 2) Will you be randomising the order of presentation of different measures, to address fatigue effects? 3) Will participants doing the survey online have the option of saving and returning to their survey later? This may help with feasibility.

- While your objectives are good/clear, I found I was wanting some clear research questions to be stated. Can I suggest that you add these, which could be potentially the research objectives reframed to key research questions, as it will make it much easier later on for the reader to see how you may address/answer these research questions.

MINOR COMMENTS

- I think in your qualitative interview a key component that would be critical to explore, that is not currently mentioned, is whether they have ever experienced any sort of mental health interventions previously (whether face to face or online). Prior experiences often influence young people's interest/motivation to seek out these sources of help - a positive experience can make it more likely; while a negative experience can deter them from seeking this sort of help again. Given the population you are looking at and your research objectives, I think it would be important to do this both in terms of asking about general experiences with mental health support, as well as cancer-specific experiences (e.g., during/since cancer treatment). Another factor that may be important to explore in the line of barriers/facilitator is what they have 'heard' about mental health support (e.g., if any of their friends has seen a psychologist), or what they heard at high school, as peer attitudes (or perceived attitudes) could be particularly persuasive in this cohort.

- While you note that you did not have any public/patient involvement in the development of the protocol, it would be good to know if you have pilot-tested your survey with any former patients at all? If you haven't, then I suggest you probably should consider it.

	This would be an easy way to check the feasibility of the survey and also assess its face validity in addressing some of the questions they may feel are most relevant when it comes to mental health help seeking?
--	---

REVIEWER	Li Ho Cheung William The University of Hong Kong, Hong Kong
REVIEW RETURNED	14-Aug-2020

GENERAL COMMENTS	The manuscript is well written and easy follow. The manuscript also addressed an important health-related issue that may have impact, especially the mental health on the cancer survivors of adolescent and young adults. Othe comments are listed below: (1) Justifications of the study. The authors argued that e-MH forAYACCS has experienced difficulties with recruitment, retention, and adherence, which require more explanation and justifications. (2) Eligibility criteria. A wide age range(16-19 years) of subjects to be recruited, which will have much variations and confounders. Some may successful completed cancer treatment for only 3 months, but some (age 39) may completed treatment 20 years! I do believe the help-seeking behaviour is rather different between adolescents and those young adults. It is unclear to me why the authors aimed to study the two different groups (adolescent and young adults). If the authors aimed to study these two groups, whether sub-group analyses will be performed to see the different help-seeking behaviour for these two groups? Statistical analyses. It comes to me that mixed approach will be used. However, the description about data analyses is very limited. Please provide more details on the data analyses, especially for the analysis of qualitative data. (3) Discussion. The discussion is rather short. More details about the potential significance of the study is needed. (4) It comes to me that ethical approval has not granted and wonder whether the protocol has been registered or not.
---

REVIEWER	Karen E. Effinger, MD, MS Emory University/Aflac Cancer and Blood Disorders Center of Children's Healthcare of Atlanta, USA
REVIEW RETURNED	26-Aug-2020

GENERAL COMMENTS	Interesting study to evaluate the factors, needs and preferences associated with mental health support in adolescent and young adult survivors of childhood cancer. The survey allows for a comprehensive evaluation of the needs of this population and the study methods allow for a robust sampling through national databases that avoid single institution biases. A few small questions arose when evaluating this study:  1. The sample size is not clearly defined in the abstract or the recruitment discussion. It would be helpful to have this stated earlier. 2. The authors mention that AYA is typically defined as 15-39; however, the study population starts at age 16. Is there a reason for this? 3. The age of research consent in many countries is 18yo. Is this different in Sweden? There is no mention of the plans for consent vs. assent for subjects aged 16-17 years. 4. I assume that number of relapses will also include an option for zero as many patients will not relapse.
---

	5. Did the study team consider also evaluating time from completion of therapy as a factor to evaluate as mental health needs may be vastly different for a participant who is 3 months off-therapy compared to someone 30 years from completion of treatment. 6. The current analysis plan does not give details of the approximate number of qualitative interviews thought to be required for data saturation and does not have details about the backward selection process. What are the cutoffs that will be employed in the backward selection? How will collinearity be addressed? 7. Are there plans to evaluate response bias through comparisons of respondents vs. non-respondents for mailed surveys? While this cannot address true non-response due to recruitment through technology-based approaches, that should be considered a study limitation.
--	---

VERSION 1 – AUTHOR RESPONSE

Reviewer 1

MAJOR COMMENTS

My major concern about this study/protocol as written is regarding the wide range of the sample – it seems that young people aged 16-39 now can participate, provided they were 0-18 years at time of diagnosis, and minimum 3 months post-treatment. So, it would seem that a young person who was an infant when they were diagnosed and is 39 now would still be eligible? This concerns me from a few perspectives: 1) I imagine that many young people particularly <5 or so years at diagnosis will not recall much of their cancer experiences; 2) you have a very large group in age who will each have experienced the disruption of cancer at different developmental stages, ranging from pre-school through to being treated for cancer during the end of high school. My questions related to these issues are as follows:

1. How are the questions phrased relating to ‘emotional distress experienced during cancer treatment’, and how will you account for young people who cannot recall this?

Thank you for raising this very important consideration. First, we chose the age period 16-39 in line with the common definition of adolescent and young adult childhood cancer survivors being between 15-39 years of age. Further, this is an age range commonly used in research when examining a wide range of physical and psychological late effects associated with childhood cancer.

In relation to the phrasing of questions pertaining to emotional distress and the cancer experience, we do not explicitly ask young adults to recall their emotional distress at the time of the cancer diagnosis and subsequent treatment. Instead, we ask young adults to think about any emotional distress related to the experience of childhood cancer more generally, for example:

Have you experienced any mental health difficulties related to your childhood cancer experience in the last 6 months?

When thinking about your childhood cancer experience, we would like you to not only think about the experience of being diagnosed and treated for cancer, but to also consider potential longer-term impacts of the experience of having childhood cancer on your life.

As such, we are not only asking young people to consider their experience of being diagnosed and treated for cancer, but also wider impacts of the cancer experience. There are a number of long-term psychological, physical, social, and developmental stressors related to the cancer experience that may negatively impact young adults’ ability to manage their psychological health that persist many years after cancer diagnosis and treatment (e.g., long-term physical and cognitive impairments, inability to work or study, reduced income, poor social support, and limitations of daily living). We have now made this clearer in the Introduction (page 3, lines 14-16):

However, significant challenges, such as late effects, remain after treatment completion.[2, 3]

Survivors of childhood cancer report a number of long-term adverse physical and developmental

outcomes [4-6] and social and economic impacts, including poorer educational and occupational achievements.[7,8]

2. Assuming that you are simply treating their cancer experience as a risk factor that may increase their need for mental health support (but not the reason for their wanting to seek support per se – i.e., not assuming that they want support specifically to address psychological issues associated with their cancer experiences) – how will you tease this apart in your questions and subsequent analyses?

Whilst we are treating young adults experience of childhood cancer as a risk factor for an increased need of mental health support, all questions in the survey pertaining to their experience of emotional health difficulties, and receipt of support, ask young adults to think about emotional health difficulties and support related specifically to their experience of childhood cancer.

In the embedded qualitative interviews, we ask young adults in more depth what type of psychological support they have accessed, and whether this has been cancer specific, or non-cancer specific. We also examine what difficulties and experiences they would like psychological interventions to target/focus. As such, we hope the embedded qualitative study will allow us to examine in more depth whether young adults would like support that specifically designed to address psychological issues associated with their cancer experience.

Further, research would suggest a need for an increased focus on the development of specific tailored support to the population, and as such, the current study is designed to help inform the future development of interventions for young adults who would specifically want support to address psychological difficulties associated with their cancer experience. In addition, evidence would suggest when developing e-MH interventions, tailoring intervention content to the specific population increases effectiveness and adherence. We have now made this clearer in the Introduction (page 5, lines 12-18):

In addition, tailoring e-MH interventions to specific patient groups has been associated with increased effectiveness [59] and adherence.[60] Indeed, rates of attrition may be reduced, and adherence increased, if the perspective of the population using the intervention is taken into account when developing e-MH interventions.[61] Further, research into the psychological needs of an AYACCS population indicates a need for increased focus on the specific psychological and psychosocial needs of the population to enable to provision of more tailored support to the cancer experience.[62] As such, there is need to explore AYACCSs general preferences for psychological support, including potential intervention content.

We have also added additional details in the discussion to emphasise that the results of the survey may lead to the development of future interventions that are specifically tailored to the needs of an AYACCS population who are seeking support in relation to their experience of cancer (page 13, lines 4-9):

Further, increasing our understanding of attitudes towards e-MH support and preferences regarding the provision of mental health support in the population may inform the development of more acceptable and relevant interventions for this underserved population. In addition, examining preferences for support will enable the development of future psychological interventions that are specifically tailored to and target the needs on an AYACCS population seeking support in relation to their experience of cancer.

3. Do your questions about childhood negative events ask about whether these occurred chronologically prior to their cancer diagnosis (as a history of e.g., school bullying prior to cancer diagnosis, for example, would predispose them to poorer social functioning/social anxiety during the

treatment period, which would in turn likely exacerbate their mental health symptoms during/after treatment)?

Thank-you for raising this important consideration. We do not ask whether childhood negative events occurred before or after their cancer diagnosis. We agree that a history of negative childhood events may predispose childhood to poorer psychosocial functioning during the treatment period. Further, as stated in the Introduction, research suggests some AYACCS may experience bullying related to their cancer experience and poor family functioning/conflict. However, given the retrospective nature of the survey, it may be challenging for some respondents to recall whether they experienced negative childhood events before or after their cancer diagnosis, dependent on age at the time. Further, we are interested more generally in whether the experience of childhood negative events may be associated with a weaker utilisation of psychological support in the population, as opposed to whether negative childhood events may predispose them to poorer psychosocial functioning and poorer mental health. Given some AYACCS may be at an increased risk of experiencing negative childhood events (e.g., bullying, family conflict) it may be helpful to understand whether this may be associated with poorer utilisation of psychological support to help improve help-seeking in the population in the future.

4. Also, will any of your items assess what issues they would want to seek help for, and whether this has been impacted in any way by their cancer experiences?

Specific issues that AYACCS would like to seek help for is not examined in the survey. However, in the embedded qualitative interview study we ask young adults what emotional difficulties they would like psychological interventions to target/focus when examining preferences for support. Further, we explore preferences in regards to intervention content areas. We have now made this clearer within the manuscript (page 11, lines 26-31):

An interview guide will be developed, consisting of open-ended questions, structured around the third research objective covering the following topics: (1) perceived need for mental health support; (2) past experience of receiving mental health support (formal/informal, cancer-experience specific/non-cancer experience specific) (3) preferences for support (e.g., type of support, emotional difficulties to be target, and intervention content areas); and (4) barriers and facilitators to help-seeking behavior (e.g., perceived attitudes of peers, knowledge/information, economic resources, geographical location, stigma).

5. Given your aims are to inform practice, how do you intend to use your analyses to do this with such a broad group? Is it the case in Sweden that the same group/hospital setting/community cancer support organisation would provide psychological support to all 16-39 year olds with some kind of history of cancer? It seems that the clinical implications for program development for teenagers with a recent history of cancer (e.g., <5 years) may be different than young adults in their 20s and 30s who were diagnosed with cancer as small children...

Thank you for highlighting this important point. The present study is designed as an exploratory study and will help to inform future research and intervention development. We do not anticipate that the study will inform an intervention designed to suit all 16-39 year olds with some kind of history of cancer, but might start to inform different interventions for different age groups. We will take into account current age in the analysis of the survey (16-24/25-39), however, we agree that these age groups are still very broad. We will also now include time since first cancer treatment (years) as a predisposing factor in our model. Further, we explore in more detail preferences for support via the embedded qualitative study, which may provide further information concerning potential implications for future intervention development. However, it is unfortunately out of scope for the current study to conduct separate embedded interview studies with different age subgroups. However, we do recognise that the broad age range included is a clear limitation, and we have now highlighted this limitation in the manuscript (page 13, lines 18-24):

Forth, the current study includes a broad age range (16-39) and it may be expected that adolescents will have differing needs and preferences to young adults in their 20s or 30s. Indeed, a common

limitation of research in the area is the inclusion of heterogeneous samples across different age groups.[97] However, the study is designed to be a first step towards developing more acceptable and relevant interventions for the population and may inform future research with specific AYACCS subgroups, taking into account a number of sociodemographic and clinical factors (e.g., gender, current age, age at diagnosis and cancer type).

6. Your survey seems very long – 98 items is quite extensive, and I would be worried about the feasibility of this and the level of burden in completing it. A few questions: 1) will participants be reimbursed at all for their time in doing the survey? If so, you need to clarify this; if not, I think this needs to be acknowledged as a limitation as I think the type of participants who are willing and able to complete such a lengthy survey for no compensation will not necessarily be representative as a group. The commitment and general self-organisation required to complete such a long survey may also preclude some people from participating precisely for the reasons that they also may have trouble organising themselves to seek mental health support, so I think this isn't irrelevant to your core study question actually. 2) Will you be randomising the order of presentation of different measures, to address fatigue effects? 3) Will participants doing the survey online have the option of saving and returning to their survey later? This may help with feasibility.

Thank-you for this comment. First, participants will not be offered any financial reimbursements for taking part in study, and we agree this may impact the generalizability of findings by potentially excluding some AYACCS subgroups. We have now acknowledged this as a limitation of the study (page 13, lines 14-18):

Third, the study does not offer any compensation to participants. Research suggests compensation can facilitate the recruitment of minority populations and those from lower socioeconomic backgrounds.[95] Subsequently, this may limit the representativeness of the sample and generalizability of results, especially to marginalized young adults who may already be at an elevated risk of not seeking mental health support.[96]

In relation to the randomization of different measures, we recognise this would be an excellent way of helping mitigate against respondent fatigue. However, randomization of the order of questionnaire presentation is a not currently a technical feature of the U-CARE-portal which will be used to support data collection in this study. However, we have now recognised this as a limitation in the manuscript (page 13, lines 24-26):

Finally, the length of the survey may lead to respondent fatigue (e.g., lower levels of attention and motivation in later sections of the questionnaire) potentially resulting in poorer quality data or missing data.

However, to help with the feasibility of participants completing the questionnaire, the survey will be open for a period of six weeks. Further, each item completed in the questionnaire is auto-saved and respondents can select "save and continue" should they wish to pause and return to the survey at a later time. To try to minimise drop out and facilitate data collection, participants will receive reminders via SMS and email to complete the online survey at 4 weeks, 2 weeks, and 2 days prior to the survey closing. We have now added these important details to the manuscript:

The online survey will be open for a period of six weeks. Each item completed in the survey is auto-saved and participants are also able to select "save and continue later" if they wish to return to the survey before submitting. (page 7, lines 35-37)

In addition, participants who consent to complete the online survey will receive reminders via SMS and email to complete the survey at 4 weeks, 2 weeks, and 2 days prior to the survey closing. (page 8, lines 14-16)

7. While your objectives are good/clear, I found I was wanting some clear research questions to be stated. Can I suggest that you add these, which could be potentially the research objectives reframed to key research questions, as it will make it much easier later on for the reader to see how you may address/answer these research questions.

Thank-you for this comment. We have now reformulated the study objectives into research questions (page 5, lines 20-36):

Study aims and research questions

The overall aims of the study are to: (1) examine potential associations between health service use factors, informed by Andersen's behavioural model of health services use, and help-seeking behaviour; (2) examine attitudes towards e-MH interventions; and (3) explore perceived need and past experiences of mental health support; preferences for support; and barriers and facilitators to help-seeking. Specifically, the following research questions will be examined:

- (1) Are there associations between predisposing, enabling, environmental, and needs related health service use factors, informed by Andersen's behavioural model of health, and help-seeking behaviour (actual help-seeking and type of help-seeking) within an AYACCS population?
- (2) What attitudes are held by an AYACCS population towards e-MH interventions?
- (3) What is the perceived need for mental health support within an AYACCS population?
- (4) What are AYACCS experiences of past receipt of cancer specific and non-cancer experience specific mental health support?
- (5) What preferences do an AYACCS population hold towards mental health support related to their cancer experience?
- (6) What are the barriers and facilitators do an AYACCSA population experience seeking help for mental health support related to their cancer experience?

8. I think in your qualitative interview a key component that would be critical to explore, that is not currently mentioned, is whether they have ever experienced any sort of mental health interventions previously (whether face to face or online). Prior experiences often influence young people's interest/motivation to seek out these sources of help - a positive experience can make it more likely; while a negative experience can deter them from seeking this sort of help again. Given the population you are looking at and your research objectives, I think it would be important to do this both in terms of asking about general experiences with mental health support, as well as cancer-specific experiences (e.g., during/since cancer treatment). Another factor that may be important to explore in the line of barriers/facilitator is what they have 'heard' about mental health support (e.g., if any of their friends has seen a psychologist), or what they heard at high school, as peer attitudes (or perceived attitudes) could be particularly persuasive in this cohort.

Thank-you for these comments.

The interview guide does currently include questions concerning the type of support received (if any) and will explore formal psychological support, and experience of formal psychological supported if received. Further, the interview will explore past experience of receiving support from family, friends, school/university, social media, and other health care professionals (e.g., family doctors). We have made the content of the interviews clearer with the manuscript, and will include specific prompts concerning cancer specific/non-cancer specific support (page 11, lines 26-31):

An interview guide will be developed, consisting of open-ended questions, structured around the third research objective covering the following topics: (1) perceived need for mental health support; (2) past experience of receiving mental health support (formal/informal, cancer-experience specific/non-cancer experience specific) (3) preferences for support (e.g., type of support, emotional difficulties to be target, and intervention content areas); and (4) barriers and facilitators to help-seeking behavior (e.g., perceived attitudes of peers, knowledge/information, economic resources, geographical location, stigma).

9. While you note that you did not have any public/patient involvement in the development of the protocol, it would be good to know if you have pilot-tested your survey with any former patients at all? If you haven't, then I suggest you probably should consider it. This would be an easy way to check the feasibility of the survey and also assess its face validity in addressing some of the questions they may feel are most relevant when it comes to mental health help seeking?

Thank-you for this suggestion. The survey has been tested by experts in clinical psychology and an intern, to examine length of time for completion, and general understanding. However, we agree that it would be beneficial to pilot-test the survey. As such, we will seek to consult with AYACCS research partners to pilot-test the survey, to examine feasibility, acceptability, and face validity of the survey – added to the public involvement section (page 12, lines 24-28):

The protocol was developed without patient and public involvement (PPI). However, we will seek to involve AYACCS in pilot-testing the survey and in data interpretation of the content analysis of the embedded semi-structured interviews. Specifically, a consultation approach to PPI will be adopted [92] in pilot-testing the survey and a panel of AYACCS research partners will be asked to complete the survey (online and paper based) and comment on the feasibility, acceptability, and relevancy of the survey.

Reviewer 2:

1. Justifications of the study. The authors argued that e-MH for AYACCS has experienced difficulties with recruitment, retention, and adherence, which require more explanation and justifications.

Thank-you for this comment. We have now added some further detail to the Introduction to justify the study (page 4, lines 4-12):

However, research examining e-MH support for an AYACCS population has encountered challenges with recruitment, attrition, and internet use.[34, 38] For example, one study examining the feasibility and acceptability of e-MH recruited only 28 participants over a 12 month period and experienced almost 30% attrition.[38] Another feasibility study of e-MH for adolescents and young adults who had experienced cancer during the adolescent period recruited only 6 participants from a total of 320 potential participants invited into the study, and experienced a 100% attrition rate.[34] Indeed, a recent review of e-Health interventions for AYACCS concluded that engaging the population was challenging, and there is currently a lack of literature concerning how to better engage the population.[36] However, challenges with recruitment, attrition, and internet use are common in the field of e-MH research.[39-41] Potentially, challenges with recruitment, attrition, and internet use may be related to help-seeking behaviour and the acceptability of e-MH for the population.

2. Eligibility criteria. A wide age range(16-39 years) of subjects to be recruited, which will have much variations and confounders. Some may successful completed cancer treatment for only 3 months, but some (age 39) may completed treatment 20 years! I do believe the help-seeking behaviour is rather different between adolescents and those young adults. It is unclear to me why the authors aimed to study the two different groups (adolescent and young adults). If the authors aimed to study these two groups, whether sub-group analyses will be performed to see the different help-seeking behaviour for these two groups?

Thank-you for raising this comment. A similar comment was raised by the first reviewer. First, we chose the age period 16-39 in line with the common definition of adolescent and young adult childhood cancer survivors being aged between 15-39 years. Further, this is an age range commonly used in research when examining a wide range of physical and psychological late effects associated with childhood cancer. However, we do expect there to be a number of differences in help seeking behaviours between adolescents and young adults. We do take into account current age in the analysis of the survey results by considering age a pre-disposing factor and we split into two subgroups (16-24/25-39) as informed by previous research (Younes et al., 2015), however, we understand these two age groups are very board. Further, we will now include time since first cancer treatment (years) as a predisposing factor in our model. We do acknowledge that the broad age range included is a clear limitation, and we have now highlighted this limitation in the manuscript (page 13, lines 18-24):

Forth, the current study includes a broad age range (16-39) and it may be expected that adolescents will have differing needs and preferences to young adults in their 20s or 30s. Indeed, a common

limitation of research in the area is the inclusion of heterogeneous samples across different age groups.[97] However, the study is designed to be a first step towards developing more acceptable and relevant interventions for the population and may inform future research with specific AYACCS subgroups, taking into account a number of sociodemographic and clinical factors (e.g., gender, current age, age at diagnosis and cancer type).

3. Statistical analyses. It comes to me that mixed approach will be used. However, the description about data analyses is very limited. Please provide more details on the data analyses, especially for the analysis of qualitative data.

Thank-you for this comment. We have now added further detail regarding both the quantitative and qualitative data analysis plan:

First, univariable analyses will be conducted for each predictor separately. Variance inflation factor will be used to examine multicollinearity prior to the regression analysis. A multivariable logistic regression will be performed using stepwise manual backward selection procedure. All predictor variables will initially be included, and non-significant predictors will be eliminated with only variables significantly related to the dependent variable ($p < 0.05$) retained. For both the univariable and multivariable logistic regressions, Odds Ratios will be presented, alongside 95% confidence intervals, and p-values (page 12, line 2-6)

An inductive content analysis approach [88] will be adopted to analyse transcriptions. Only manifest content will be analysed and the following steps will be undertaken: (1) each interview transcript will be read multiple times; (2) meaning units will be identified; (3) meaning units will be labelled used descriptive codes; (4) codes will be sorted into themes and subthemes.[88] (page 12, lines 12-16)

4. Discussion. The discussion is rather short. More details about the potential significance of the study is needed.

Thank-you for this comment. We have added some additional details to the discussion, including the potential significance of the study, and some limitations raised by reviewers (page 12, lines 39-40, page 13, lines 1-30):

Our study design allows us to collect both qualitative and quantitative data and therefore will provide a more in-depth and rich understanding of mental health help-seeking behaviour and attitudes towards e-MH support in an AYACCS population. The identification of factors associated with help-seeking behaviour may help identify potential targets to improve help-seeking behaviour in the future. For example, interventions could be developed to target factors found to be negatively associated with help-seeking [94] and subsequently enhance help-seeking behaviours in an AYACCS population in the future. Further, increasing our understanding of attitudes towards e-MH support and preferences regarding the provision of mental health support in the population may inform the development of more acceptable and relevant interventions for this underserved population. In addition, examining preferences for support will enable the development of future psychological interventions that are specifically tailored to and target the needs on an AYACCS population seeking support in relation to their experience of cancer.

Despite the strengths of this study, the design presents some limitations. . First, only those AYACCS who report receiving mental health support in the past six months are defined as help-seeking. However, this approach may lead to the exclusion of AYACCS who have attempted to seek formal help but not accessed. Second, the cross-sectional design does not allow us to draw any conclusions regarding cause and effect. Third, the study does not offer any compensation to participants. Research suggests compensation can facilitate the recruitment of minority populations and those from lower socioeconomic backgrounds.[95] Subsequently, this may limit the representativeness of the sample and generalizability of results, especially to marginalized young adults who may already be at an elevated risk of not seeking mental health support.[96] Forth, the current study includes a broad

age range (16-39) and it may be expected that adolescents will have differing needs and preferences to young adults in their 20s or 30s. Indeed, a common limitation of research in the area is the inclusion of heterogeneous samples across different age groups.[97] However, the study is designed to be a first step towards developing more acceptable and relevant interventions for the population and may inform future research with specific AYACCS subgroups, taking into account a number of sociodemographic and clinical factors (e.g., gender, current age, age at diagnosis and cancer type). Fifth, the length of the survey may lead to respondent fatigue (e.g., lower levels of attention and motivation in later sections of the questionnaire) potentially resulting in poorer quality data or missing data. Finally, we will not evaluate response bias via comparisons of respondents versus non-responders for mailed surveys.

In conclusion, results of the study may have the potential to improve access to tailored, relevant, and acceptable mental health support for this currently underserved population.

5. It comes to me that ethical approval has not been granted and wonder whether the protocol has been registered or not.

We are still currently in the process of applying for ethical approval for the study from the Swedish Ethical Review Authority and hope to receive ethical approval soon. We have not registered the study in any study registry as this is not a clinical trial, systematic review, or observational study that assess health outcomes. However, if an appropriate study registry could be recommended we would be very pleased to register the study protocol.

Reviewer 3:

1. The sample size is not clearly defined in the abstract or the recruitment discussion. It would be helpful to have this stated earlier.

Thank-you for this comment. We have now moved the "Sample size estimation" subsection under Recruitment and have added details to be the Abstract (page 2, line 12):

Methods and analysis: An online and paper-based cross-sectional self-report survey (98 items) and embedded qualitative interview study across Sweden, with a target sample size of n=365.

3. The authors mention that AYA is typically defined as 15-39; however, the study population starts at age 16. Is there a reason for this?

Thank you for this question. Whilst AYA is typically defined as 15-39, to include participants under 16 years of age would involve the need for parental consent from both legal guardians, as well as assent from the participant. As such, a decision was made to not include 15 year olds due to the complexities of including a double consent procedure (adolescent assent and consent from both legal guardians) for this age group.

4. The age of research consent in many countries is 18yo. Is this different in Sweden? There is no mention of the plans for consent vs. assent for subjects aged 16-17 years.

Thank-you for this comment. Generally, the age of research consent in Sweden is 18 years. However, adolescent minor consent (e.g., consent from participants aged 16 and 17) can be sought and approved for studies that examine potentially sensitive topics with adolescents, especially when risk is minimal (e.g., survey studies). The current study explores topics that may be considered as sensitive, and given common difficulties experienced by the population, may involve discussions concerning mental health, fertility, sexual health, and bullying. As such, asking for parental consent may deter some adolescents from participation. Further, many adolescents have the capacity to understand important informational elements in a research study in a manner similar to adults (Roth-Cline & Nelson, 2013). As such, we will ask for consent, rather than assent, for participants aged 16-17 years.

5. I assume that number of relapses will also include an option for zero as many patients will not relapse.

Yes. This survey item first asks whether the participant has experienced a relapse. If yes, we ask whether one, or more than one, have been experienced. We have now made this clearer in the manuscript (page 8, line 36-37):

(2) number of relapses (zero, one or more than one);

6. Did the study team consider also evaluating time from completion of therapy as a factor to evaluate as mental health needs may be vastly different for a participant who is 3 months off-therapy compared to someone 30 years from completion of treatment.

Thank-you for this important comment. We are now including time since first cancer treatment (years) as a predisposing factor in our model (see Table 1). In addition, as raised in response to other reviewer comments, we have included this as a limitation in the manuscript:

Forth, the current study includes a broad age range (16-39) and it may be expected that adolescents will have differing needs and preferences to young adults in their 20s or 30s. Indeed, a common limitation of research in the area is the inclusion of heterogeneous samples across different age groups.[97] However, the study is designed to be a first step towards developing more acceptable and relevant interventions for the population and may inform future research with specific AYACCS subgroups, taking into account a number of sociodemographic and clinical factors (e.g., gender, current age, age at diagnosis and cancer type).

7. The current analysis plan does not give details of the approximate number of qualitative interviews thought to be required for data saturation and does not have details about the backward selection process. What are the cutoffs that will be employed in the backward selection? How will collinearity be addressed?

Thank-you for these comments. We have now added further detail to the quantitative data analysis plan in the manuscript, which are highlighted below (page 12, line 2-6):

First, univariable analyses will be conducted for each predictor separately. Variance inflation factor will be used to examine multicollinearity prior to the regression analysis. A multivariable logistic regression will be performed using stepwise manual backward selection procedure. All predictor variables will initially be included, and non-significant predictors will be eliminated with only variables significantly related to the dependent variable ($p < 0.05$) retained. For both the univariable and multivariable logistic regressions, Odds Ratios will be presented, alongside 95% confidence intervals, and p-values

In relation, to the number of qualitative interviews required for data saturation, this is very difficult to predict and is a decision made during the analytical process. However, based on other research, we may anticipate needing to interview approximately 20 – 30 participants. We have now included this detail in the manuscript (page 11, lines 18-21):

Respondents meeting these criteria will be invited consecutively until thematic data saturation is met.[85, 86] As the decision concerning whether data saturation has been met is made during the analytic process it is difficult to determine the number of interviews a priori, however we may anticipate interviewing approximately 20-30 participants.[87]

8. Are there plans to evaluate response bias through comparisons of respondents vs. non-respondents for mailed surveys? While this cannot address true non-response due to recruitment through technology-based approaches, that should be considered a study limitation.

Thank-you for this comment. There are no plans to evaluate response bias through comparisons of respondents versus non-respondents for mailed surveys to AYACCS identified via the childhood cancer registry. Unfortunately, we have not included the analysis of the basic demographic data provided by the Childhood Cancer Registry in our analysis plan, however, we understand this is a limitation: (page 13, line 26-27):

Finally, we will not evaluate response bias via comparisons of respondents versus non-responders for mailed surveys.

VERSION 2 – REVIEW

REVIEWER	William Li The University of Hong Kong Hong Kong SAR
REVIEW RETURNED	16-Nov-2020

GENERAL COMMENTS	The authors have addressed most of the comments. The authors explained some major issues and presented them as limitations instead of revising the protocol. We well understand it is difficult to modify a funded protocol. However, differentiated from completed research, the authors still have opportunities to avoid the limitations or design defects before the implementation. The author is strongly recommended to consider the reviewers' suggestion. Please find more details below:  1. As stated, this study included a population with a wide age range and many variations. Following the study aims, stepwise regression may not sufficient to provide reliable results. Subgroup analysis, lasso regression can be considered in the data analysis plan. In addition, the dependent variable, which is the key to the regression, is not described clearly in the analysis section. 2. This study should adopt consistent definitions in the protocol. In the eligible inclusion criteria, participants will be aged 16-39 yrs, which the author has fully explained in the responses. However, those were diagnosed with cancer when 0-18 years, which the author defined it as a period of Childhood. The inconsistency of definition may lead to complicated situations for the enrolled participants. It is suggested to include the participants who will be aged 16-39 yrs, and have been diagnosed with cancer when 0-16 yrs, or who will be aged 18-39 yrs, and have been diagnosed with cancer when 0-18 yrs. 3. This study adopts a mixed-method, including observation investigations and interviews, which are included in the broad definition of a clinical trial. Hence, clinical trial registration is strongly recommended. 4. The author defined the help-seeking as receiving mental health support, which was contradictory. In addition, one of the aims in this study is to explore the barriers experienced by the targeted population to the help seek. The experience of those who failed to receive support will be more important to the development of the intervention. It will be a loss if they are excluded.
---

REVIEWER	Ursula Sansom-Daly University of New South Wales (UNSW), Sydney, Australia
REVIEW RETURNED	24-Nov-2020

GENERAL COMMENTS	The authors have done a very thorough job of addressing all of my comments and I think the additional detail has strengthened the paper. I have no further concerns at this stage -well done.
---

VERSION 2 – AUTHOR RESPONSE

Reviewer: 2

Comments to the Author

The authors have addressed most of the comments. The authors explained some major issues and presented them as limitations instead of revising the protocol. We well understand it is difficult to modify a funded protocol. However, differentiated from completed research, the authors still have opportunities to avoid the limitations or design defects before the implementation. The author is strongly recommended to consider the reviewers' suggestion. Please find more details below:

Thank-you for this comment. We very much appreciate your review and the opportunity affords us to improve the design/overcome limitations before we commence the study. We have tried to address your comments as best we can. However, there are some limitations we cannot overcome. For example, we cannot easily increase the sample size given this is a PhD study with restrictions concerning time and funding. As such, we may not be powered to conduct the subgroup analysis we have now planned based on your recommendations below. Further, given time restrictions, we cannot make major changes to the protocol that will require further review by the Ethical Review Board or are significantly different to the funded research plan. Despite these restrictions, we have tried to incorporate your specific suggestions below.

1. As stated, this study included a population with a wide age range and many variations. Following the study aims, stepwise regression may not sufficient to provide reliable results. Subgroup analysis, lasso regression can be considered in the data analysis plan. In addition, the dependent variable, which is the key to the regression, is not described clearly in the analysis section.

Thank-you for this comment, we understand we have heterogeneous study population. We have now added subgroup analysis concerning age and gender. However, the planned subgroup analysis will be exploratory and we are not able to consider in a revised sample size calculation (page 12, lines 3-7):

Given the heterogeneous nature of the study population, subgroup analysis will be performed by stratifying the population by age and sex. The same method of modelling will be used for the subgroup analyses for age (15-19 years vs. 20-29 years vs. 30-39 years) and sex (male vs female). The subgroup analysis will be exploratory and with results used to inform subsequent studies.

We have made the dependent variable clearer in the data analysis plan (page 22, lines 34-37):

The potential association between predicting factors as informed by Andersen's behavioural model of health services use (predisposing, enabling, environmental, and need-related factors) and the dependent variable help-seeking behaviour e.g., receipt of mental health support in the past 6 months will be examined using univariable and multivariable logistic regressions.[70]

2. This study should adopt consistent definitions in the protocol. In the eligible inclusion criteria, participants will be aged 16-39 yrs, which the author has fully explained in the responses. However, those were diagnosed with cancer when 0-18 years, which the author defined it as a period of Childhood. The inconsistency of definition may lead to complicated situations for the enrolled participants. It is suggested to include the participants who will be aged 16-39 yrs, and have been

diagnosed with cancer when 0-16 yrs, or who will be aged 18-39 yrs, and have been diagnosed with cancer when 0-18 yrs.

The period 0-18 years is consistent with the childhood cancer period we have used within our previous research (e.g., Wikman et al., 2018; Woodford et al., 2018) and is used by other large studies in the area, for example, the St. Jude Lifetime Cohort Study (Bhakta et al., 2017). The age period 15-39 years is commonly used to define adolescent and young adult childhood cancer survivors: who are the target study population. However, as mentioned in the previous response letter, AYACCS aged 15 years will not be included due to the need for parental consent.

It is not unusual for studies within this field to recruit cancer survivors diagnosed during the childhood period (e.g., diagnosed aged 0–18 years), who at the time of study participation are within an age range that overlaps with the potential age at cancer diagnosis (e.g., Cheung et al., 2017; Ruiz et al., 2016). This is due to the age range commonly used to define childhood cancer partially overlapping with the age range commonly used to define adolescence and young adulthood in a childhood cancer survivor population.

Given that we use both a common definition of childhood cancer and a common definition of adolescent and young adulthood, we do not feel that our inclusion criteria is inconsistent and thus we do not feel comfortable making the suggested changes. However we have made some amendments to the manuscript, which we hope will aid readers in understanding the study population (page 3, lines 23-26):

As such, adolescent and young adult childhood cancer survivors (AYACCS), commonly defined as cancer survivors diagnosed during childhood currently aged between 15-39 years,[18] are particularly vulnerable to experiencing adverse psychological and social outcomes, and there is a significant need for long-term care and support.[5]

Eligible participants will: (1) be an adolescent or young adult, aged 16-39 years, at study start; (2) have been diagnosed with childhood cancer when 0-18 years;

Bhakta N, Liu Q, Ness KK, et al. The cumulative burden of surviving childhood cancer: an initial report from the St Jude Lifetime Cohort Study (SJLIFE). *Lancet*. 2017;390(10112):2569-2582

Cheung CK, Zebrack B. What do adolescents and young adults want from cancer resources? Insights from a Delphi panel of AYA patients. *Support Care Cancer*. 2017 25(1):119-126

Ruiz ME, Sender L, Torno L, Fortier MA. The Associations of Age and Ethnicity on Substance Use Behaviors of Adolescent and Young Adult Childhood Cancer Survivors. *Psychooncology*. 2016 25(10):1229-1236

Wikman A, Mattson E, von Essen L, Hovén E. Prevalence and predictors of symptoms of anxiety and depression, and comorbid symptoms of distress in parents of childhood cancer survivors and bereaved parents five years after end of treatment or a child's death. *Acta Oncologica*. 2018 57(7):950-957

Woodford J, Wikman A, Cernvall M, Ljungman G, Romppala A, Grönqvist H, von Essen L. Study protocol for a feasibility study of an internet-administered, guided, CBT-based, self-help intervention (ENGAGE) for parents of children previously treated for cancer. *BMJ Open*. 2018 8(6):e023708

3. This study adopts a mixed-method, including observation investigations and interviews, which are included in the broad definition of a clinical trial. Hence, clinical trial registration is strongly recommended.

Thank-you for this recommendation. We have now registered the study in the ISRCTN registry: ISRCTN70570236.

The ISRCTN number has been added to the manuscript:

Page 2, line 25: Trial registration number: ISRCTN70570236

Page 6, line 2: An online and paper-based, cross-sectional, self-report survey and embedded qualitative interview study across Sweden. The study has been registered in the ISRCTN clinical trial registry (ISRCTN70570236).

4. The author defined the help-seeking as receiving mental health support, which was contradictory. In addition, one of the aims in this study is to explore the barriers experienced by the targeted population to the help seek. The experience of those who failed to receive support will be more important to the development of the intervention. It will be a loss if they are excluded.

Thank-you for this comment. We realise that there is some lack of clarity in the manuscript.

Within the embedded qualitative interview study, one of our main aims is to examine barriers and facilitators to help-seeking in the population. Whilst we will only interview participants with current and/or past experience of mental health difficulties, participants do not have to be help-seekers. We have made this clearer in the manuscript (page 11, lines 14-15):

As we are interested in both barriers and facilitators to seeking help, we will endeavour to interview participants classified as help-seekers and non-help-seekers.

We have also identified that we did not make it clear in the analysis plan that the non-help seeking group will act as a reference group, as informed by similar studies in the help-seeking literature (Boerema et al., 2016). Further, any identified negative associations with help-seeking and specific factors in Andersen's behavioural model of health services use, may help us identify potential barriers to seeking help in the population. For example if stigma is found to be negatively associated with help-seeking in the population, this may indicate that we need to try and reduce stigma (e.g., education programmes) in future intervention research. We have tried to make this clearer in the manuscript:

Page 11, lines 34-38: The potential association between predicting factors as informed by Andersen's behavioural model of health services use (predisposing, enabling, environmental, and need-related factors) and the dependent variable help-seeking behaviour e.g., receipt of mental health support in the past 6 months will be examined using univariable and multivariable logistic regressions.[70] The non-help-seeking group will be used as the reference group.[70]

Page 13, lines 1-45: The identification of factors both positively and negatively associated with help-seeking behaviour may help identify potential targets to improve help-seeking behaviour in the future. For example, interventions could be developed to target and overcome factors found to be negatively associated with help-seeking [92] and subsequently enhance help-seeking behaviours in an AYACCS population in the future.

Boerema AM, Kleiboer A, Beekman AT et al. Determinants of help-seeking behavior in depression: a cross-sectional study. *BMC Psychiatry* 2016;16. doi:10.1186/s12888-016-0790-0.

Reviewer: 1

Comments to the Author

The authors have done a very thorough job of addressing all of my comments and I think the additional detail has strengthened the paper. I have no further concerns at this stage -well done.

Thank-you for your comment. We are pleased that you feel we have thoroughly addressed your comments and subsequently strengthened the manuscript and study.